**Data Availability Statement:** Data from the Austrian Centre for Statistics and Documentation in Intensive Care (ASDI) used to generate the dataset

# Association of immediate versus delayed extubation of patients admitted to intensive care units postoperatively and outcomes: A retrospective study

Paul Zajic [1]*, Michael Eichinger[1], Michael Eichlseder [1], Barbara Hallmann[1], Gabriel Honnef [1], Tobias Fellinger[2], Barbara Metnitz[3], Martin Posch[2], Martin Rief [1], Philipp G. H. Metnitz[1]

1 Division of General Anaesthesiology, Emergency- and Intensive Care Medicine, Medical University of Graz, Graz, Austria, 2 Center for Medical Statistics, Informatics and Intelligent Systems, Medical University of Vienna, Vienna, Austria, 3 Austrian Center for Documentation and Quality Assurance in Intensive Care, Vienna, Austria

* paul.zajic@medunigraz.at

## Abstract

### Aim of this study

This study seeks to investigate, whether extubation of tracheally intubated patients admitted to intensive care units (ICU) postoperatively either immediately at the day of admission (day 1) or delayed at the first postoperative day (day 2) is associated with differences in outcomes.

### Materials and methods

We performed a retrospective analysis of data from an Austrian ICU registry. Adult patients admitted between January 1st, 2012 and December 31st, 2019 following elective and emergency surgery, who were intubated at the day 1 and were extubated at day 1 or day 2, were included. We performed logistic regression analyses for in-hospital mortality and over-sedation or agitation following extubation.

### Results

52 982 patients constituted the main study population. 1 231 (3.3%) patients extubated at day 1 and 958 (5.9%) at day 2 died in hospital, 464 (1.3%) patients extubated at day 1 and 613 (3.8%) at day 2 demonstrated agitation or over-sedation after extubation during ICU stay; OR (95% CI) for in-hospital mortality were OR 1.17 (1.01–1.35, p = 0.031) and OR 2.15 (1.75–2.65, p<0.001) for agitation or over-sedation.

### Conclusions

We conclude that immediate extubation as soon as deemed feasible by clinicians is associated with favourable outcomes and may thus be considered preferable in tracheally intubated patients admitted to ICU postoperatively.

analysed in this study are subject to contractual provisions between ASDI and contributing intensive care units, hospitals, or hospital trusts, respectively; these legal stipulations do not allow for publication of any data outside defined intents of use (i.e., aggregation of data for statistical analyses and quality assurance). The dataset underlying this study can thus be obtained via the corresponding author or from the database holding organisation (ASDI, Langenzersdorferstrasse 28, A-1210, Vienna, Austria, office@asdi.ac.at) upon reasonable request after legal review.

**Funding:** The author(s) received no specific funding for this work.

**Competing interests:** The authors have declared that no competing interests exist.

## Introduction

Following surgical procedures and anaesthesia, patients may be admitted to intensive care units (ICU) with an endotracheal tube in place for several reasons. These encompass perceived or potential airway compromise, manifest or possible respiratory insufficiency, hemodynamic instability and the need for further resuscitation, continuing effects of anaesthetics and neuromuscular blocking agents, and profound hypothermia [1–3].

Patients with persistent disturbances in their vital functions may require intubation and mechanical ventilation for extended periods of time. Contrarily, patients who remain tracheally intubated postoperatively for more rapidly reversible reasons may be extubated promptly following admission to intensive care and correction of underlying causes. While guidance on extubation timing has been published [4], individual decisions have to be made by health care professionals.

This decision-making process will usually mostly be driven by current patient physiology and the perceived safety of extubation [5]. Other factors, such as reported or expected difficult airway management [6], assumed need for further stabilization, presumed complexity of post-extubation care, concurrent staff workload, and even time of day or night [7,8] may also influence extubation timing in patients ready for extubation soon after admission to intensive care. Overall, too little is known about possible influences of extubation timing on patient-oriented outcomes.

Some author groups have demonstrated, that extubation as early as in the operating room is feasible even in patients who have undergone cardiac surgery [9] and lung resection [10], others have shown that early extubation in intensive care units is associated with shortened lengths of ICU stay after aortic surgery [10–12]. Prolonged duration of surgery and general anaesthesia have been demonstrated to be associated with the occurrence of delirium [13]. Conversely, extubation failure has been demonstrated to be associated with adverse outcomes [14,15]. Clinicians might therefore choose to postpone extubation to a later timepoint perceived as optimal.

Based on these previous findings, we hypothesize that delay of extubation until the first postoperative day may lead to unwanted patient-oriented outcomes, such as increased length of stay in both ICU and hospital, increased rates of hyperactive or hypoactive delirium, and even in-hospital mortality.

### Aim of this study

This study seeks to investigate, whether extubation at either the day of ICU admission (day 1) or at the first postoperative day (day 2) of tracheally intubated patients admitted to intensive care units postoperatively is associated with in-hospital mortality, rate of re-intubation, ICU length of stay, and occurrence of agitation or over-sedation, irrespective of illness severity.

## Materials and methods

The ethics committee of the Medical University of Graz (IRB00002556) approved of the study at February 5th, 2021 (decision number 33–202 ex 20/21). The need for written informed consent was waived by ethics committee, since no study-related interventions were performed on human subjects and data used could not be traced back to individual patients, rendering them anonymous.

### Study design and setting

This study was conducted as a retrospective analysis of data collected by the Austrian Center for Documentation and Quality Assurance in Intensive Care Medicine (ASDI). Core contents of its database and variable definitions were described in detail in previous publications [16].

In brevity, documented data encompass: sociodemographic data (age, sex, and chronic conditions), the reason for ICU admission, severity of illness (measured by Simplified Acute Physiology Score 3 (SAPS 3) [17]) at ICU admission, intensity of provided care (measured by Simplified Therapeutic Intervention Scoring System (TISS-28) [18]) per day of ICU stay, degree of sedation and agitation (measured by the Riker Sedation and Agitation Scale (SAS) [19]) per day of ICU stay, administrative data (length of ICU and hospital stay), and outcome information (survival status at ICU and hospital discharge).

This manuscript adheres to the applicable STROBE (STrengthening the Reporting of OBservational studies in Epidemiology) guidelines [20].

## Patient population

Adult patients (age ≥ 18 years at ICU admission) admitted to ICUs contributing data to the ASDI registry between January 1st, 2012 and December 31st, 2019 following all surgical procedures (elective and emergency procedures) were included into this study, if information on vital status at hospital discharge and SAPS 3 score were complete. Data concerning patients not tracheally intubated at ICU admission were excluded from the study.

## Conduct of the study

The actual population of interest were patients in whom immediate extubation was or would have been feasible. Since decision processes made by treating health care professionals could not be reproduced from the available data, patients not extubated at day 1 or day 2 were excluded to form a population of patients extubated at day 1 or day 2 (= *main population*).

This approach was thought to be at risk for bias, since patients, who would have been planned for extubation at day 1 or day 2, but died or were otherwise lost earlier (e.g., discharged from the hospital), would theoretically not be considered. The potential bias introduced was assumed to favour extubation at day 2.

To reduce the risk of this bias, a subgroup of the main population was formed, which encompassed only patients admitted to the ICU until the second postoperative day or longer (= *curtailed population*). Comparison of in-hospital mortality in this curtailed population was assumed to favour extubation at day 1.

In order to further address potential bias, a target trial design, that emulated an experiment designed to randomly assign patients ready to be extubated to day 1 extubation or day 2 extubation, was additionally used [21,22].

An artificial cohort of patients predicted to be extubated at day 1 or day 2 was formed using logistic LASSO regression with the individual components of the SAPS 3 score as covariates and day 1 extubation or day 2 extubation versus any later day as the dependent variable. Patients, whose predicted probability of extubation at day 1 or day 2 exceeded a threshold of 80%, were included in this cohort (= *target trial population*). In this population, outcomes of patients extubated at day 1 versus patients extubated at any later day were compared.

## Primary and secondary endpoints

Primary outcome of interest for this study was in-hospital mortality. Secondary endpoints were the need for tracheal reintubation during ICU stay, ICU length of stay, and occurrence of agitation or over-sedation during ICU stay after extubation.

## Measurements and data handling

Information on tracheal intubation was derived from TISS-28 data. As per national specification, every day's most invasive installations or most demanding procedures were recorded by health care professionals. Patients were considered tracheally intubated at days, where a tracheal tube was documented as an airway device. Extubation was deduced to have occurred at the day before the first day without a tracheal tube in place. Tracheal reintubation was considered to have occurred, if a tracheal tube was documented as an airway installation after at least one day without a tracheal tube in place.

Admission time was derived from administrative data documented in the registry and grouped in blocks of eight hours each (00:00–07:59, 08:00–15:59, 16:00–23:59) to best reflect common shift patterns. Lengths of ICU and hospital stay, expressed in days, were calculated as the timespan between ICU admission and ICU discharge or hospital admission and hospital discharge, respectively.

SAPS 3 score was used to adjust for severity of illness. Acuteness of the surgical procedure leading to ICU admission–elective or unscheduled–and type of surgical procedure–abdominal/transplantation, trauma/orthopaedic, neurosurgical, cardiothoracic, and others–were also derived from SAPS 3 data.

Agitation and sedation as surrogates for delirium were inferred from SAS scores. Primarily, over-sedation was defined as SAS values below three ("very sedated") after extubation, agitation was defined as SAS values above five ("very agitated"). An alternative, more inclusive, definition for agitation and over-sedation, defined as SAS values below or above four ("sedated" and "agitated", respectively) after extubation, was also used. Unfavourable mental status in general was assumed, if any of these conditions was met after extubation.

## Statistical analysis

Unless stated otherwise, descriptive results and between-group comparisons are expressed as median and interquartile range (IQR) or number (n) and percentage (%).

To address the endpoints of in-hospital mortality and re-intubation, logistic regression analysis models for in-hospital mortality or reintubation, respectively, as the dependent variable were fitted. These models were adjusted for day 1 extubation or day 2 extubation and other relevant variables of adjustment as described above, i.e., SAPS 3 score, time of day, acuteness of surgery, and type of surgery.

To assess ICU length of stay, multivariable Cox cause-specific hazard regression analysis with discharge from the ICU as the dependent variable was performed; models were also adjusted for day 1 extubation or day 2 extubation as well as SAPS 3 score, time of day, acuteness of surgery, and type of surgery. Cases of patients who deceased before discharge from the ICU were censored at their respective times of death.

For the endpoint of agitation or over-sedation after extubation, logistic regression analysis models for the composite endpoint of the occurrence of either as dependent variables were fitted. These models were adjusted for day 1 extubation or day 2 extubation and possible published predictive factors of delirium [23,24], insofar as they were available in the registry: age, SAPS 3 score, serum creatinine, alcoholism, time of day, acuteness of surgical procedure, and type of surgical procedure. Additionally, ordinal regression analysis, adjusted for the aforementioned covariables, was performed to assess associations of variables of interest with stepwise deviations of SAS from 4 ("calm and cooperative").

All models were further adjusted for anonymous identifiers of the treating intensive care units as fixed effects; these results were not depicted. All statistical analyses were performed using R version 4.1.2.

## Results

### Patient populations

The initial query yielded 325 442 patient data sets from 154 intensive care units (Fig 1). Of these patients, 77 515 were tracheally intubated at ICU admission, 52 982 of whom were extubated at day 1 or day 2 and thus constituted the main population. 420 patients died or were discharged from hospital within three days of ICU admission and were thus excluded to form the curtailed population of 52 562. The target trial population was made up of 33 990 data sets of patients, of whom 31 027 (91%) were extubated at day 1 or day 2.

Patients in the main cohort were mostly male (58%), had a median age of 67 (IQR 56–76) years, and were mostly admitted to ICU following elective surgery (71%). 36 811 patients were extubated at day 1 and 16 171 were extubated at day 2. Median length of ICU stay was 3 (IQR

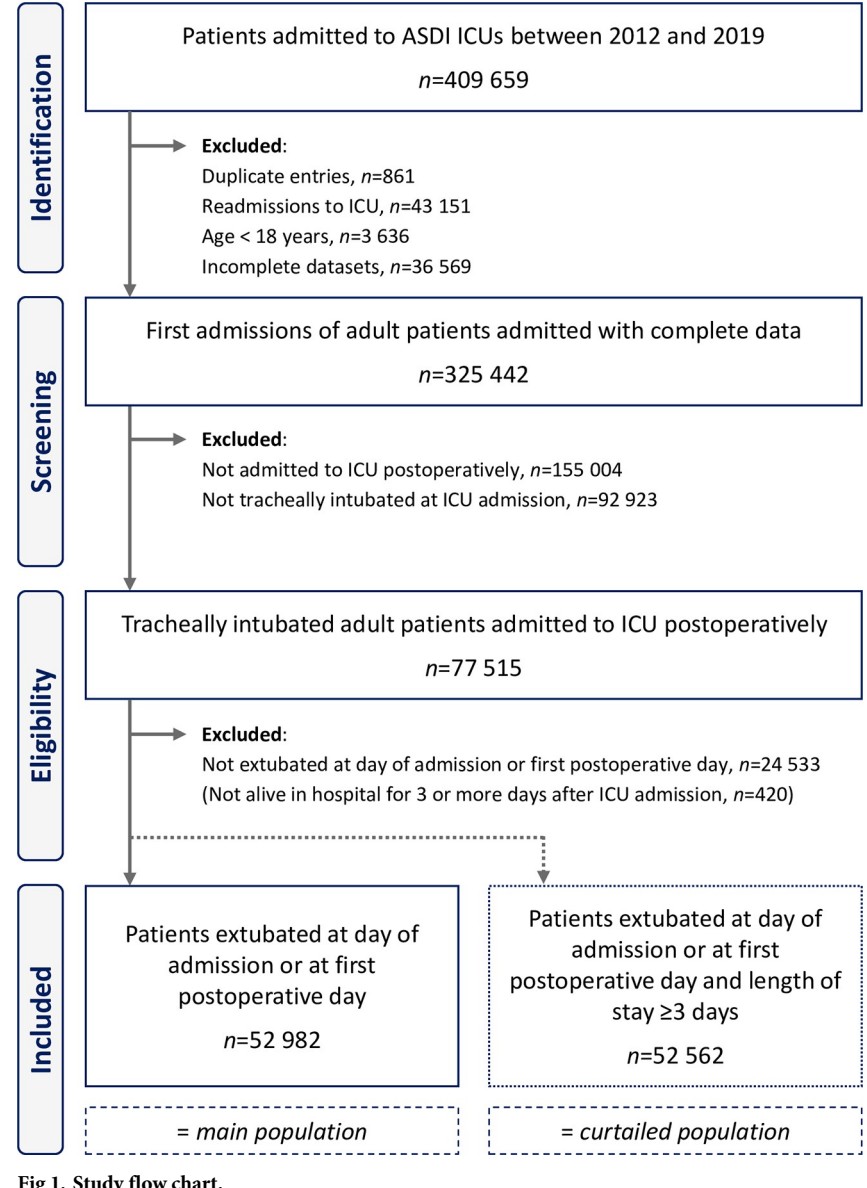

**Fig 1. Study flow chart.**

**Table 1. Baseline patient characteristics, characteristics of surgery, and outcomes in the main patient population.**

| | | Extubation at | |
|---|---|---|---|
| | Overall | Day 1 (day of ICU admission) | Day 2 (1st postoperative day) |
| *n* of patients | 52982 | 36811 | 16171 |
| Age [years] (median, IQR) | 67 (56–76) | 67 (55–75) | 69 (57–76) |
| Male sex (n, %) | 30871 (58%) | 21115 (57%) | 9756 (60%) |
| Type of surgery (n, %) | | | |
| Abdominal / transplant | 13748 (26%) | 9819 (27%) | 3929 (24%) |
| Cardiothoracic | 18683 (35%) | 12593 (34%) | 6090 (38%) |
| Neuro | 8676 (17%) | 7105 (19%) | 1571 (10%) |
| Trauma / orthopaedic | 3230 (6%) | 1960 (5%) | 1270 (8%) |
| Other | 8645 (16%) | 5334 (15%) | 3311 (20%) |
| Urgency of surgery (n, %) | | | |
| Elective surgery | 37789 (71%) | 28250 (77%) | 9539 (59%) |
| Emergency surgery | 12807 (24%) | 6938 (19%) | 5869 (36%) |
| Unspecified | 2386 (5%) | 1623 (4%) | 763 (5%) |
| SAPS 3 score (median, IQR) | 43 (36–52) | 42 (35–50) | 46 (39–55) |
| Time of ICU admission (n, %) | | | |
| 00:00–07:59 | 4281 (8%) | 3392 (9%) | 889 (5%) |
| 08:00–15:59 | 37850 (71%) | 28685 (78%) | 9165 (57%) |
| 16:00–23:59 | 10851 (21%) | 4734 (13%) | 6117 (38%) |
| ICU length of stay [days] (median, IQR) | 3 (2–4) | 2 (2–4) | 3 (2–5) |
| Hospital length of stay [days] (median, IQR) | 15 (10–22) | 14 (10–21) | 16 (11–26) |
| Re-intubation during ICU stay (n, %) | 1968 (4%) | 1151 (3%) | 817 (5%) |
| In-hospital mortality (n, %) | 2189 (4%) | 1231 (3%) | 958 (6%) |
| Agitation or over-sedation during ICU stay after extubation | | | |
| SAS>5 or SAS<3 | 1077 (2%) | 464 (1%) | 613 (4%) |
| SAS>4 or SAS<4 | 4470 (8%) | 1854 (5%) | 2616 (16%) |

ICU = intensive care unit, IQR = inter-quartile range, SAPS 3 = Simplified Acute Physiology Score 3, SAS = Riker Sedation and Agitation Scale.

2–4) days, median length of hospital stay was 15 (IQR 10–22) days (Table 1). Patient characteristics in the curtailed population were very similar overall (S1 Table in S1 File). Patient characteristics in the target trial population were slightly different, as patients were predominantly admitted following elective surgery (93%) (S2 Table in S1 File).

## Main population

In the main population, 1 231 (3.3%) of patients extubated at day 1 and 958 (5.9%) of patients extubated at day 2 died in hospital (Table 1). Multivariable logistic regression analysis with good predictive performance (AUC = 0.817) identified day 2 extubation compared to day 1 extubation as an independent risk factor for in-hospital mortality (OR 1.17, 95% CI 1.01–1.35, p = 0.031) (Table 2).

1151 (3.1%) patients extubated at day 1 and 817 (5.1%) extubated at day 2 required re-intubation during ICU stay (Table 1). In multivariable logistic regression analysis with adequate predictive performance (AUC = 0.756), day 2 extubation compared to day 1 extubation was a significant risk factor for re-intubation (OR 1.22, 95% CI 1.05–1.42, p = 0.002) (S3 **Table in S1 File**).

Median length of ICU stay was 2 (IQR 2–4) days in day 1 extubation and 3 (IQR 2–5) days in day 2 extubation, respectively. Multivariable Cox cause-specific hazard regression analysis

**Table 2. Multivariate logistic regression analysis for in-hospital mortality as the dependent variable and variables of interest and adjustment as co-variates.** The model includes anonymous ICU identifiers as fixed effects, these are not depicted.

| Variable | OR | 95% CI | p |
|---|---|---|---|
| Extubation at | | | |
| Day 1 (day of ICU admission) | 1.00 | | |
| Day 2 (1st postoperative day) | 1.17 | 1.01–1.35 | 0.031 |
| SAPS 3 score | 1.07 | 1.07–1.08 | <0.001 |
| Type of surgery | | | |
| Abdominal / transplant | 1.00 | | |
| Cardiothoracic | 0.60 | 0.47–0.75 | <0.001 |
| Neuro | 0.34 | 0.24–0.48 | <0.001 |
| Trauma / orthopaedic | 1.07 | 0.84–1.37 | 0.998 |
| Other | 0.60 | 0.49–0.74 | <0.001 |
| Urgency of surgery | | | |
| Elective surgery | 0.95 | 0.80–1.13 | 0.995 |
| Emergency surgery | 1.00 | | |
| Unspecified | 1.62 | 1.02–2.57 | 0.032 |
| Time of ICU admission | | | |
| 00:00–07:59 | 1.00 | | |
| 08:00–15:59 | 1.01 | 0.80–1.28 | >0.999 |
| 16:00–23:59 | 1.03 | 0.81–1.32 | >0.999 |

AUC = 0.817. CI = confidence interval, ICU = intensive care unit, OR = odds ratio, SAPS 3 = Simplified Acute Physiology Score 3.

found day 2 extubation to be associated with a lower hazard of ICU discharge (HR 0.66, 95% CI 0.60–0.73, p = 0.002) (S4Table in S1 File). Median length of hospital stay was 14 (IQR 10–21) days in day 1 extubation and 16 (11–26) days in day 2 extubation.

464 (1.3%) and 613 (3.8%) patients following day 1 extubation and day 2 extubation, respectively, demonstrated either agitation or over-sedation after extubation at least once during ICU stay (Table 1). Multivariable logistic regression analysis with adequate predictive performance (AUC = 0.796) identified day 2 extubation compared to day 1 extubation as a significant risk factor (OR 2.15, 95% CI 1.75–2.65, p<0.001) (Table 3). Findings were similar when the analysis was repeated using the more inclusive definition of SAS not equal to 4 (S5 **Table in S1 File**). Ordinal regression analysis also found day 2 extubation compared to day 1 extubation to be significant risk factor for deviations from normal SAS values (OR 3.22, 95% CI 2.78–3.73, p<0.001) (S6 **Table in S1 File**).

## Curtailed population

Unadjusted outcomes of interest in the curtailed population were very similar to the main population (S1 **Table in S1 File**). Multivariable logistic regression analysis models also demonstrated day 2 extubation compared to day 1 extubation to be associated with higher risk of in-hospital mortality (OR 1.23, 95% CI 1.06–1.43, p = 0.001)(S7 **Table in S1 File**), re-intubation (OR 1.20, 95% CI 1.03–1.39, p = 0.008)(S8 **Table in S1 File**), and agitation or over-sedation during ICU stay (OR 2.12, 95% CI 1.72–2.62, p<0.001)(S9 **Table in S1 File**). Multivariable Cox cause-specific hazard regression analysis also found day 2 extubation to be associated with a lower hazard of ICU discharge (HR 0.67, 95% CI 0.61–0.74, p = 0.002) (S10 **Table in S1 File**).

**Table 3. Multivariate logistic regression analysis for agitation (SAS>5) or over-sedation (SAS<3) during ICU stay after extubation as the dependent variable and variables of interest and adjustment as co-variates.** The model includes anonymous ICU identifiers as fixed effects, these are not depicted.

| Variable | OR | 95% CI | p |
|---|---|---|---|
| Extubation at | | | |
| Day 1 (day of ICU admission) | 1.00 | | |
| Day 2 (1st postoperative day) | 2.15 | 1.74–2.65 | <0.001 |
| SAPS 3 score | 1.02 | 1.01–1.03 | <0.001 |
| Age [years] | 1.02 | 1.01–1.02 | <0.001 |
| Serum creatinine at ICU admission [mg/dl] | 1.06 | 0.98–1.14 | 0.277 |
| Alcoholism | 1.90 | 1.20–3.01 | 0.001 |
| Type of surgery | | | |
| Abdominal / transplant | 1.00 | | |
| Cardiothoracic | 1.33 | 0.94–1.90 | 0.209 |
| Neuro | 1.27 | 0.79–2.04 | 0.852 |
| Trauma / orthopaedic | 1.67 | 1.14–2.46 | 0.002 |
| Other | 0.91 | 0.64–1.30 | 0.999 |
| Urgency of surgery | | | |
| Elective surgery | 0.74 | 0.57–0.98 | 0.023 |
| Emergency surgery | 1.00 | | |
| Unspecified | 1.30 | 0.76–2.24 | 0.885 |
| Time of ICU admission | | | |
| 00:00–07:59 | 1.00 | | |
| 08:00–15:59 | 0.76 | 0.54–1.07 | 0.239 |
| 16:00–23:59 | 0.97 | 0.68–1.39 | >0.999 |

AUC = 0.796. CI = confidence interval, ICU = intensive care unit, OR = odds ratio, SAPS 3 = Simplified Acute Physiology Score 3, SAS = Riker Sedation and Agitation Scale.

## Target trial population

In the target trial population, 426 (1.7%) of patients extubated at the day of ICU admission and 619 (6.5%) of patients extubated at day 2 or later died in hospital (S2 **Table in S1 File**). Multivariable logistic regression analysis found later extubation compared to day 1 extubation to be an independent risk factor for in-hospital mortality (OR 3.33, 95% CI 2.70–4.10, p<0.001) (S11 **Table in S1 File**), re-intubation (OR 1.74, 95% CI 1.40–2.18, p<0.001)(S12 **Table in S1 File**), and agitation or over-sedation during ICU stay (OR 10.84, 95% CI 8.54–13.76, p<0.001) (S13 **Table in S1 File**).

## Discussion

In this large, retrospective, registry-based study we demonstrate that immediate extubation–i.e., extubation at the day of ICU admission–is associated with better outcomes compared to delayed extubation–i.e., extubation at the first post-operative day–in tracheally intubated patients admitted to intensive care units after surgery, who do not require longer-term ventilatory support, irrespective of the underlying severity of illness. This encompasses lower risk of in-hospital mortality, and lower rates of agitation or over-sedation during ICU stay.

Overall, in-hospital mortality in this cohort was low when compared to general ICU populations. The found in-hospital mortality rate closely resembled that previously described for non-cardiac surgery in general [25], but was lower than that in a prior study in a similar cohort

[26]. Nevertheless, a statistically significant and clinically relevant difference in risk-adjusted in-hospital mortality was found between patients extubated at the day of ICU admission and those extubated at the following day.

Similarly, the rate of agitation or over-sedation states was found to be comparably low in this patient cohort. That may at least partly be due to the primarily used definition, which required patients to either be "very agitated" or "very sedated". This rather restrictive definition was chosen to avoid retrospective over-classification of patients based on the available scale, which was primarily developed to assess disturbances in attention and awareness, but less so memory, orientation, or perception. [18] Still, a statistically significant and clinically relevant difference in baseline-risk adjusted odds of agitation or over-sedation occurrence was found between patients extubated at the day of ICU admission and those extubated at the following day. This finding is supported by models using a more inclusive definition of unfavourable mental status.

Neither acuity of surgery nor time of admission to the ICU were associated with risk of in-hospital mortality. It is uncertain, whether surgery at or until night time is associated with worse outcomes than operations during the day. A recent systematic review and meta-analysis suggested that night/after-hours surgery may be associated with a higher risk of mortality based on low-certainty evidence [27]. Contrarily, a large Europe-wide study could not find any relationship between the time of day at which the procedure was performed and mortality risk [28].

Previous, smaller studies were mainly conducted in patients admitted to ICUs after coronary artery bypass grafting. A randomized controlled trial of early extubation (between 1 and 6 hours) later extubation (between 12 to 22 hours) concluded that early extubation does not increase perioperative morbidity, but reduces ICU and hospital lengths of stay [29]. Retrospective studies found patients intubated for shorter periods of time were at lower risk of death [30], had shorter stays in both the ICU and the hospital [30,31]. and were at lower risk for the development of delirium [30,32].

Findings of this study underline the importance of swift tracheal extubation in the ICU of patients admitted postoperatively. A decisive approach to termination of sedation and extubation and patients who do not require ventilatory support beyond the first postoperative day is associated with lower risk of death, unwanted mental state, and prolonged ICU stay when compared to more deferred approaches.

## Weaknesses and strengths

This study was retrospective in nature and therefore was subject to all limitations of retrospective analyses, although all data used in this study were collected prospectively. Reasons for decisions made by treating clinicians could therefore not necessarily be represented to their full extent, which may introduce unknown bias not adjusted for in entirety. Delays in extubation might have occurred due to physiologic reasons not entirely adjusted for by the SAPS 3 score. Contrarily, conduction of a randomized controlled trial would at least be ethically challenging, as it would require delayed extubation of patients already deemed fit for Extubation.

Multiple approaches to reduce bias, including a target trial design [21,22]. were employed to mitigate the risk of bias. The target trial population, however, might not include all relevant variables, and therefore, might not correctly specify the population (i.e., the inclusion/exclusion criteria) that would be included in a randomized controlled trial.

The sizeable underlying database allowed for analyses of a large patient cohort resembling everyday clinical practice. It was, however, somewhat restricted in detail and granularity; therefore, timing of extubation could only be reported per day. This in turns may have led to

episodes of tracheal reintubation on the day of extubation being overlooked. Time passed between ICU admission and extubation may also have varied due to different times of ICU admission; however, ICU admission time was considered as a factor in all multivariable analyses.

Delirium could only be defined using the outer boundaries of the Riker Sedation and Agitation scale documented once per day. While this scoring system was a well-established method of mental status in intensive care units, more specific tools such as the Confusion Assessment Method in Intensive Care Units (CAM-ICU) [33] or the Intensive Care Delirium Screening Checklist (ICDSC) [34] would be more sensitive in the detection of delirium states. Predictive power of our models was less optimal for agitation or over-sedation, although variables of adjustment were carefully chosen based on the available literature.

## Conclusions

Delayed extubation of tracheally intubated patients admitted to intensive care units for postoperative care is associated with adverse patient-oriented outcomes. Randomized controlled trials are warranted to further explore the matter, if they are found ethically feasible. At this time, clinicians may strive for immediate extubation in patients ready for extubation as soon as deemed feasible and safe.

## Supporting information

**S1 Checklist. STROBE statement—Checklist of items that should be included in reports of observational studies.**
(PDF)

**S1 File. PDF supplementary materials including S1 Table S1 to S13 Tables.**
(PDF)

## Acknowledgments

We thank all physicians, nurses and allied health care professionals working in Austrian intensive care units contributing to the ASDI benchmarking and quality assurance project.

## Author Contributions

**Conceptualization:** Paul Zajic, Barbara Hallmann, Tobias Fellinger, Martin Rief, Philipp G. H. Metnitz.

**Data curation:** Tobias Fellinger, Martin Posch.

**Formal analysis:** Paul Zajic, Tobias Fellinger, Barbara Metnitz, Martin Posch.

**Investigation:** Paul Zajic, Michael Eichinger, Michael Eichlseder, Barbara Hallmann, Gabriel Honnef, Tobias Fellinger, Martin Posch, Martin Rief, Philipp G. H. Metnitz.

**Methodology:** Paul Zajic, Tobias Fellinger, Barbara Metnitz, Martin Posch, Philipp G. H. Metnitz.

**Project administration:** Paul Zajic, Martin Posch.

**Resources:** Martin Posch, Philipp G. H. Metnitz.

**Software:** Tobias Fellinger.

**Supervision:** Barbara Metnitz, Martin Posch, Philipp G. H. Metnitz.

**Validation:** Michael Eichlseder, Barbara Hallmann, Gabriel Honnef, Barbara Metnitz, Martin Rief, Philipp G. H. Metnitz.

**Writing – original draft:** Paul Zajic, Michael Eichlseder.

**Writing – review & editing:** Paul Zajic, Michael Eichinger, Barbara Hallmann, Gabriel Honnef, Martin Rief, Philipp G. H. Metnitz.

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
