## [Decision Letter · Decision Letter 0]

14 Oct 2022

PONE-D-22-23243Association of immediate versus delayed extubation of patients admitted to intensive care units postoperatively and outcomes: a retrospective studyPLOS ONE

Dear Dr. Zajic,

Thank you for submitting your manuscript to PLOS ONE. After careful consideration, we feel that it has merit but does not fully meet PLOS ONE’s publication criteria as it currently stands. Therefore, we invite you to submit a revised version of the manuscript that addresses the points raised during the review process.

The manuscript has merit but the opinions of the Reviewers was contrasting. Moreover, I agree that some aspects of the analysis must be clarified. Specifically, the definition of "curtailed population" must be better defined. The authors should evaluate the possibility to perform an analysis using ICU lenght of stay. Regarding the confounders that have been suggested, the authors should clearly insert in the discussion, as a major limitations, the lack of these data for their models (if they are not able to provide them).

We look forward to receiving your revised manuscript.

Kind regards,

Andrea Cortegiani, M.D.

Academic Editor

PLOS ONE

Journal Requirements:

Additional Editor Comments:

The manuscript has merit but the opinions of the Reviewers was contrasting. Moreover, I agree that some aspects of the analysis must be clarified. Specifically, the definition of "curtailed population" must be better defined. The authors should evaluate the possibility to perform an analysis using ICU lenght of stay. Regarding the confounders that have been suggested, the authors should clearly insert in the discussion, as a major limitations, the lack of these data for their models (if they are not able to provide them).

Reviewers' comments:

Reviewer's Responses to Questions

**Comments to the Author**

1. Is the manuscript technically sound, and do the data support the conclusions?

Reviewer #1: Yes

Reviewer #2: No

2. Has the statistical analysis been performed appropriately and rigorously? 

Reviewer #1: Yes

Reviewer #2: Yes

3. Have the authors made all data underlying the findings in their manuscript fully available?

Reviewer #1: Yes

Reviewer #2: No

4. Is the manuscript presented in an intelligible fashion and written in standard English?

Reviewer #1: Yes

Reviewer #2: Yes

5. Review Comments to the Author

Reviewer #1: Dear Editor and Authors,

Thank you for the opportunity to revise the manuscript titled “Association of immediate versus delayed extubation of patients admitted to intensive care units postoperatively and outcomes: a retrospective study”.

The manuscript aims to investigate whether immediate vs 24 hours delayed extubation of post-surgical patients admitted to ICU can influence in-hospital mortality, incidence of reintubation during ICU stay and occurrence of agitation or over-sedation during ICU stay after extubation.

The design of the study is retrospective, based on an Austrian registry (the Austrian Center for Documentation and Quality Assurance in Intensive Care Medicine).

The authors found that immediate extubation is associated with favorable outcomes.

The article is well written, methods are clearly stated, and results and discussion are well expressed and argued.

In my opinion, only minor revision is needed:

- Pag 3 line 60: the authors give an overview of the several factors that may influence the decision-making process of proceeding to extubation upon which staff workload and time of the day citing a retrospective cohort study. On this matter I would suggest a recent meta-analysis DOI: 10.1097/EJA.0000000000001579.

- Pag 4 line 98: the authors state that the article adheres to the STROBE guidelines. I suggest the authors to cite the literature related to these guidelines and, if possible, to add as supplementary material the related checklist.

Reviewer #2: Dear authors,

thank you for pulling out this important work. To my mind however, there are many issues that need to be adressed before further consideration.

Major issues

1) I could not understand what the authors wanted to highlight about the risk of bias (P5 L112) and the definition of the curtailed population???

2) Definition of Day 1 and day2. As this is the primary endpoint, this point deserves more details in the methods. I believe that the day of extubation was defined according to the time table in the data-base, meaning that a patient admitted at 11:00PM and extubated and 1:00AM was counting as Day-2? It seems poorly relevant since a patient admitted at 5:00AM and extubated at 10:00PM would clearly display a longer duration of invasive mechanical ventilation than the first one, but would be identified as a Day-1 extubation.

3) Mortality as a primary endpoint; although the authors have taken into account several confounders I find it hard to believe in this message. I believe that there are many in-hospital confounders that cannot be taken into account with the present database (infection, second look surgery, medical postoperative complications...). I believe that an ICU-centred outcome could be more relevant (ICU length of stay for instance).

4) As the duration of the study goes from 2012 to 2017 it would be relevant to see whether practices about extubation have changed over time

5) Can you provide statistical univariate analysis for Table 1 and others? There seems to be more urgent surgery in the day-2 surgery which is a mjor confounder for both mortality and the decision for extubation.

6) Can the authors describe what is the Target trial? I mean did you perform a Propensity score analysis and how was it performed?

6. PLOS authors have the option to publish the peer review history of their article (what does this mean?). If published, this will include your full peer review and any attached files.

Reviewer #1: No

Reviewer #2: No

---

## [Author Response · Author response to Decision Letter 0]

16 Dec 2022

Editor

"The manuscript has merit but the opinions of the Reviewers was contrasting. Moreover, I agree that some aspects of the analysis must be clarified. Specifically, the definition of "curtailed population" must be better defined. The authors should evaluate the possibility to perform an analysis using ICU lenght of stay. Regarding the confounders that have been suggested, the authors should clearly insert in the discussion, as a major limitations, the lack of these data for their models (if they are not able to provide them)."

We thank the editor for consideration of our manuscript. Throughout this document, we respond to points raised by reviewers and outline changes and additions made to the manuscript and its supplementary materials in order to comply with requests by the editor and reviewers.

Reviewer #1

"The article is well written, methods are clearly stated, and results and discussion are well expressed and argued."

We thank the reviewer for their mindful assessment of our manuscript and their kind words.

"- Pag 3 line 60: the authors give an overview of the several factors that may influence the decision-making process of proceeding to extubation upon which staff workload and time of the day citing a retrospective cohort study. On this matter I would suggest a recent meta-analysis DOI: 10.1097/EJA.0000000000001579."

We thank the reviewer for their valuable suggestion and have added the above-mentioned meta-analysis to the references in our manuscript.

"- Pag 4 line 98: the authors state that the article adheres to the STROBE guidelines. I suggest the authors to cite the literature related to these guidelines and, if possible, to add as supplementary material the related checklist."

We are again grateful for this suggestion for improvement and have cited the STROBE guidance document (as published in PLOS Med) in our manuscript and have added the filled-in STROBE checklist to this submission’s documents.

Reviewer #2

"thank you for pulling out this important work. To my mind however, there are many issues that need to be adressed before further consideration."

We thank the reviewer for their thorough assessment of our manuscript, seek to address the fair points brought up by the reviewer, and present changes made in order to improve the manuscript.

"

"1) I could not understand what the authors wanted to highlight about the risk of bias (P5 L112) and the definition of the curtailed population???

Since risk of bias is certainly an important issue, we thank the reviewer for their plea for clarification. In a theoretical (albeit hardly feasible, as pointed out in the manuscript) randomised controlled trial on the subject, patients would have to be randomised at the time of admission to ICU. In such a design, patients who would decease unexpectedly or would be discharged/transferred (however unlikely that may be) before the intervention (i.e., extubation) had been performed, could be included and assessed. In a retrospective study design (that we deem necessary) based on the inclusion and exclusion criteria, such and assessment is not possible (although the target-trial design discussed later seeks to emulate that). This puts the analysis at risk for immortal time bias (Hernán M et al. J Clin Epidemiol. 2016 doi: 10.1016/j.jclinepi.2016.04.014). This is the reason for the “curtailed population”. We have emphasised that issue in the manuscript accordingly. 

"2) Definition of Day 1 and day2. As this is the primary endpoint, this point deserves more details in the methods. I believe that the day of extubation was defined according to the time table in the data-base, meaning that a patient admitted at 11:00PM and extubated and 1:00AM was counting as Day-2? It seems poorly relevant since a patient admitted at 5:00AM and extubated at 10:00PM would clearly display a longer duration of invasive mechanical ventilation than the first one, but would be identified as a Day-1 extubation."

We thank the reviewer for this insightful comment. The observed lack of granularity of the underlying dataset is indeed a limitation of our study, which has previously been discussed in the “Weaknesses and strengths” section of our manuscript. To ameliorate this issue, we have included “Time of ICU admission” as a covariable in our descriptive and regression analyses throughout our study. In doing so, we demonstrate, that night-time admission (00:00 – 07:59) is exceedingly rare in our patient cohort, which makes these shortened timeframes unlikely, and we adjust for that factor.

"3) Mortality as a primary endpoint; although the authors have taken into account several confounders I find it hard to believe in this message. I believe that there are many in-hospital confounders that cannot be taken into account with the present database (infection, second look surgery, medical postoperative complications...). I believe that an ICU-centred outcome could be more relevant (ICU length of stay for instance)."

We agree with the reviewer that endpoint selection is undoubtedly essential for study design and validity. Hospital mortality was chosen as the primary outcome of interest firstly because it is a patient-centred metric of indisputable relevance for patients and treating clinicians alike and secondly because the Simplified Acute Physiology Score 3 (SAPS 3) was derived for prediction of hospital mortality (Moreno RP et al. Intensive Care Med. 2005. doi: 10.1007/s00134-005-2763-5). While influence factors (both positive and negative) during hospital stay and treatment certainly influence outcomes, risk prediction based on well-calibrated scores such as the SAPS 3 documented at ICU admission is known to produce reliable results, as highlighted by the AUC of 0.82 in this study’s main population. 

We already report on ICU-centred outcomes such as ICU length of stay, however, these are very prone to interference from organisation or practice and are thus suboptimal as primary outcomes. Nevertheless, we have added multivariable Cox cause-specific hazard regression analysis models for discharge from the ICU as the dependent variable as additional analyses; these are cited in the manuscript and depicted as tables S4 and S10 in the supplementary materials. 

Please note: table legends of other tables have changed due to the incorporation of additional analyses to allow for chronologic ordering of the supplementary materials.

"4) As the duration of the study goes from 2012 to 2017 it would be relevant to see whether practices about extubation have changed over time"

We agree with the reviewer’s assertation that changes over time in policy and management seem possible. As pointed out in the manuscript, no definitive guidance exists either nationally; it is therefore not possible to demonstrate changes (or the lack thereof) of practice guidance over time. With regards to internal validity, though, no notable changes have taken place within the units participating to this dataset; we have added a table describing frequencies of day 1 and day 2 extubation, respectively, to the reviewer materials as table R1. 

"5) Can you provide statistical univariate analysis for Table 1 and others? There seems to be more urgent surgery in the day-2 surgery which is a mjor confounder for both mortality and the decision for extubation."

As per the reviewer’s request, we have attached univariate analyses (both including and not-including ICU-IDs as fixed effects) for all co-variables used in our main model as reviewer materials to this response; please see tables R2 to R11 in the reviewer materials.

"6) Can the authors describe what is the Target trial? I mean did you perform a Propensity score analysis and how was it performed?"

We are delighted for the reviewer’s interest in the target trial approach used in our study. As pointed out in the manuscript, the “target-trial population” was formed using logistic regression analysis with LASSO (least absolute shrinkage and selection operator). Individual components of the SAPS 3 score documented at ICU admission (e.g., vital parameters, reasons for admission, pre-existing disease) were used as covariates and day 1 extubation or day 2 extubation versus any later day were used as the dependent variable. Patients who had a calculated chance of extubation at day 1 of 80% or more were included into the target trial population.

---

## [Decision Letter · Decision Letter 1]

10 Jan 2023

Association of immediate versus delayed extubation of patients admitted to intensive care units postoperatively and outcomes: a retrospective study

PONE-D-22-23243R1

Dear Dr. Zajic,

We’re pleased to inform you that your manuscript has been judged scientifically suitable for publication and will be formally accepted for publication once it meets all outstanding technical requirements.

Kind regards,

Andrea Cortegiani, M.D.

Academic Editor

PLOS ONE

Additional Editor Comments (optional):

Reviewers' comments:

Reviewer's Responses to Questions

**Comments to the Author**

1. If the authors have adequately addressed your comments raised in a previous round of review and you feel that this manuscript is now acceptable for publication, you may indicate that here to bypass the “Comments to the Author” section, enter your conflict of interest statement in the “Confidential to Editor” section, and submit your "Accept" recommendation.

Reviewer #1: All comments have been addressed

Reviewer #2: All comments have been addressed

2. Is the manuscript technically sound, and do the data support the conclusions?

Reviewer #1: Yes

Reviewer #2: Yes

3. Has the statistical analysis been performed appropriately and rigorously? 

Reviewer #1: Yes

Reviewer #2: Yes

4. Have the authors made all data underlying the findings in their manuscript fully available?

Reviewer #1: Yes

Reviewer #2: No

5. Is the manuscript presented in an intelligible fashion and written in standard English?

Reviewer #1: Yes

Reviewer #2: Yes

6. Review Comments to the Author

Reviewer #1: The authors answered acceptably the concerns reported on the first review.

They sought to improve references and to provide checklist in the supplementary material.

Reviewer #2: Dear authors thank you for answering all my questions. I believe that all issues have been adressed and I do not have further comments.

7. PLOS authors have the option to publish the peer review history of their article (what does this mean?). If published, this will include your full peer review and any attached files.

Reviewer #1: No

Reviewer #2: No

---

## [Editor Report · Acceptance letter]

13 Jan 2023

PONE-D-22-23243R1 

Association of immediate versus delayed extubation of patients admitted to intensive care units postoperatively and outcomes: a retrospective study 

Dear Dr. Zajic:

I'm pleased to inform you that your manuscript has been deemed suitable for publication in PLOS ONE. Congratulations! Your manuscript is now with our production department. 

Kind regards, 

on behalf of

Dr. Andrea Cortegiani 

Academic Editor

PLOS ONE